# Silver Nanoparticle-Functionalised Nitrogen-Doped Carbon Quantum Dots for the Highly Efficient Determination of Uric Acid

**DOI:** 10.3390/molecules27144586

**Published:** 2022-07-19

**Authors:** Qianchun Zhang, Shuxin Du, Fengling Tian, Xixi Long, Siqi Xie, Shan Tang, Linchun Bao

**Affiliations:** Key Laboratory for Analytical Science of Food and Environment Pollution of Qian Xi Nan, School of Biology and Chemistry, Minzu Normal University of Xingyi, Xingyi 562400, China; dusuxin@xynun.edu.cn (S.D.); tianfengling@xynun.edu.cn (F.T.); longxixi@xynun.edu.cn (X.L.); tangshan@xynun.edu.cn (S.T.); linchunbao1985@163.com (L.B.)

**Keywords:** AgNPs/N-CQDs, fluorescence detection, uric acid, serum sample

## Abstract

The fabrication of efficient fluorescent probes that possess an excellent sensitivity and selectivity for uric acid is highly desirable and challenging. In this study, composites of silver nanoparticles (AgNPs) wrapped with nitrogen-doped carbon quantum dots (N-CQDs) were synthesised utilising N-CQDs as the reducing and stabilising agents in a single reaction with AgNO_3_. The morphology and structure, absorption properties, functional groups, and fluorescence properties were characterised by transmission electron microscopy, X-ray photoelectron spectroscopy, Fourier transform infrared spectroscopy, ultraviolet spectroscopy, fluorescence spectroscopy, and X-ray diffraction spectroscopy. In addition, we developed a novel method based on AgNPs/N-CQDs for the detection of uric acid using the enzymatic reaction of uric acid oxidase. The fluorescence enhancement of the AgNPs/N-CQDs composite was linear (*R*^2^ = 0.9971) in the range of 2.0–60 μmol/L, and gave a detection limit of 0.53 μmol/L. Trace uric acid was successfully determined in real serum samples from the serum of 10 healthy candidates and 10 gout patients, and the results were consistent with those recorded by Qianxinan Prefecture People’s Hospital. These results indicate that the developed AgNP/N-CQD system can provide a universal platform for detecting the multispecies ratio fluorescence of H_2_O_2_ generation in other biological systems.

## 1. Introduction

Uric acid (UA), the final product of purine metabolism, is considered a valuable diagnostic biomarker of several diseases [1]. The excessive accumulation of UA leads to diseases such as gout, hyperuricemia, neurological, cardiovascular, and Lesh–Nyhan syndrome, among others. In particular, hyperuricemia is usually defined as a human serum UA level > 0.46 mM [2,3,4], while a low UA concentration has been reported as a risk factor for multiple sclerosis and oxidative stress [5]. Therefore, it is vital to develop simple, fast, and efficient methods for monitoring UA levels during early clinical diagnosis and treatment processes. Previously, researchers have devised a variety of analytical techniques to detect UA in human physiological samples, such as conventional assays based on the enzymatic method [6], gas chromatography [7], high-performance liquid chromatography (HPLC) [8], electrochemical assays [9], and fluorescence assays [10]. Among these methods, gas chromatography, HPLC, and electrochemical methods often involve complex sample preparation and separation processes, are time consuming and can require surface modification. In contrast, fluorescence spectroscopy is a favourable method for detecting UA because of its low cost, simple operation, fast response time, good selectivity, and high sensitivity. To develop novel analytical approaches for sophisticated instrumental devices, new probe materials for techniques such as fluorescence spectroscopy are required. To date, a range of fluorescent probes has been exploited based on organic molecules [11], metal nanoclusters [12], and carbon quantum dots (CQDs) [13] and their composites [14]. Among them, the detection of UA using CQDs wrapped with silver nanoparticles has great potential [15]. CQDs are biocompatible zero-dimensional carbon-based materials with unique physicochemical properties, and as a result, they are commonly used for fluorescence sensing and bioimaging [16,17]. It has been shown that the doping of relevant atoms, such as nitrogen atoms, into CQDs, can improve their optical properties and render them more applicable [18]. Moreover, N-CQDs bearing large numbers of hydroxyl groups on their surfaces can provide good growth conditions for silver nanoparticles (AgNPs), eventually forming composite AgNPs/N-CQDs with controlled sizes and a uniform distribution, without the requirement for additional reducing agents or stabilisers. In addition, since the surface plasmon resonance bands of AgNPs in the visible region overlap with the fluorescence emission spectra of N-CQDs, such composites have potential for use in the fluorescence detection of certain small molecules and toxic compounds [19,20]. The development of a novel AgNPs/N-CQDs nanocomposite for UA determination is therefore of particular interest.

Thus, we herein report the development of a novel AgNPs/N-CQDs nanocomposite as a fluorescent probe for the detection of UA. In particular, hydroxyethyl cellulose and L-citrulline were firstly used to synthesize the N-CQDs via a hydrothermal method. Because the hydroxyethyl cellulose has abundant -OH functional groups, the as-prepared N-CQDs are facilely coordinating with silver ions. Moreover, the AgNPs present in the AgNPs/N-CQDs will be oxidised to Ag^+^ by H_2_O_2_ to reduce absorption of the light emitted from the N-CQDs and enhance the fluorescence intensity of the nanocomposite. A novel detection method based on the H_2_O_2_-mediated fluorescence enhancement of the AgNPs/N-CQDs is then developed to detect UA using the enzymatic catalytic oxidation reaction, and this method is applied to the UA detection of human serum. The formation and application of these AgNPs/N-CQDs are outlined in Figure 1.

## 2. Results and Discussion

### 2.1. Characterisation of the AgNP/N-CQDs

The transmission electron microscopy (TEM) image shown in Figure 1A reveals that the prepared N-CQDs consist of well-dispersed spherical nanocomposites. HAADF (High Angle Annular Dark Field)-scanning transmission electron microscopy (STEM) observations were then performed to study the structural features of the prepared AgNPs/N-CQDs. As shown in Figure 1B,C, a clear brightness contrast can be observed for the composites, namely, a brighter centre and a darker edge. This can be attributed to the fact that the atomic numbers of C, N, and O (i.e., the elements present in the N-CQDs) are close to one another, while the atomic number of Ag (i.e., constituting the AgNPs) is significantly higher. More specifically, the intensity of the HAADF image is proportional to the atomic number [21], and so the brighter centre corresponds to the AgNPs while the darker edges correspond to the N-CQDs. Energy dispersive spectroscopy (EDS) line scanning was also used to explore the C and Ag species. As shown in Figure 1D, at a distance of 10 nm from the particle, no apparent energy absorption is observed for Ag; although, the energy absorption of C remained stable between the full scan range examined herein (i.e., 0–50 nm). Upon moving closer to the particle and over the particle (i.e., 10–45 nm positions), the energy absorption of Ag becomes higher than that of C, before returning to the zero state again. These results agree with the HRTEM image shown in Figure 1E, wherein the amorphous structure at the bottom of the AgNPs/N-CQDs is comparable to that of the N-CQDs presented in Figure 1A. Moreover, in Figure 1E, the evident lattice characteristics of Ag can be observed by the naked eye, and there is an obvious interface between the AgNPs and the N-CQDs. These results indicate that the AgNPs successfully grow on the N-CQDs to form the desired composites (AgNPs/N-CQDs).

Furthermore, the X-ray diffraction (XRD) results presented in Figure 2A show a broad peak at ~23°, which is consistent with the (002) crystal plane of graphite, and was attributed to the N-CQDs. In addition, the diffraction peaks observed at 38.1, 44.3, 64.4, and 77.1° corresponded to the (111), (200), (220), and (311) planes of Ag (PDF # 87-0717) [22]. Moreover, the Fourier transform infrared (FTIR) spectrum (Figure 2B) shows absorption bands at 2998–3601 and 2919 cm^−1^, which were attributed to the O–H/N–H and C–H stretching vibrations, respectively. The band corresponding to the C=O bending vibrations appeared at 1645.12 cm^−1^, and peaks at 1570, 1328, and 1024 cm^−1^ were attributed to the N–H, C–N, and C–O–C functionalities, respectively [23,24]. X-ray photoelectron spectroscopy (XPS) was then applied to determine the surface chemical composition of the AgNP/N-CQDs, and to illustrate the mechanism of UA detection by the composite. As shown in Figure 2C, three main peaks were observed at 400.0, 369.9, and 531.6 eV in the XPS survey spectrum, which corresponded to N 1s, Ag 3d, and O 1s, respectively. In addition, the peaks at 399.5 and 401.0 eV correlate with the bonding structures of the C–N and N–H groups, respectively (Figure 3A). The N 1s peak can also be resolved into two components at 399.2 and 400.5 eV (Figure 3B). Furthermore, the peak of 367.7 eV was attributed to the bonding structure of the Ag 3d_5/2_ bonds, while the peak located at 374.0 eV was assigned to the bonding structure of the Ag 3d_3/2_ bonds, thereby indicating that metallic silver was present in the composite material [25,26] (Figure 3C). Moreover, the O 1s peak was composed of three components at 530.9, 531.7, and 532.8 eV, which corresponded to the adsorbed oxygen species C=O, C–OH, and C–O–C (Figure 3E), respectively [27,28]. Figure 3B,D,F show the XPS patterns of the AgNPs/N-CQD composite after its reaction with H_2_O_2_ (200 μM). Compared with the pattern observed before the reaction, the intensity of the peak corresponding to the C–N bond decreased in the N 1s spectrum, while that of the N–H bond increased. Furthermore, the intensity of the peak corresponding to the Ag–N bond in the Ag 3d spectrum was enhanced [29], thereby indicating that the enhanced fluorescence of the AgNPs/N-CQD composite originated from the formation of Ag–N in the reaction system, which is a stable coordination structure.

The fluorescence properties of the N-CQDs and the AgNPs/N-CQDs were then examined using fluorescence and ultraviolet spectroscopy. The fluorescence excitation and emission matrices of the N-CQDs and the AgNPs/N-CQDs were compared, and it was observed that the fluorescence intensity of the AgNPs/N-CQDs composite was significantly lower than that of the N-CQDs (Figure 4A,C) due to the fact that the ultraviolet absorption of the AgNPs/N-CQDs (426 nm) overlaps with the fluorescence emission of N-CQDs (428 nm) (Figure 4B). This produces a fluorescence inner filtration effect (fluorescence resonance energy transfer) between the N-CQDs and the AgNPs, ultimately resulting in a decrease in the fluorescence emission intensity of the AgNPs/N-CQDs. In addition, Figure 4B shows a narrow band at 400–495 nm in the emission spectrum, which is likely due to the relatively narrow size distribution of the resultant AgNPs/N-CQDs. The optical properties of the prepared AgNPs/N-CQDs were therefore further investigated, and the fluorescence spectra are depicted in Figure 4D, where the maximum excitation wavelength was observed at 340 nm and the maximum emission wavelength was found at 408 nm.

### 2.2. Selectivity of the Assay

The selectivity of the AgNPs/N-CQD-based fluorescence probe system was then the evaluated system for the detection of UA in biological samples. Several potential interferents (5000 μmol/L), including four amino acids (L-tyrosine, L-cysteine, L-phenylalanine (L-pll), and L-lysine), glucose, AA, K^+^, Na^+^, Ca^2+^, Fe^3+^, and Mg^2+^, were introduced to investigate the fluorescence response of the AgNPs/N-CQDs for comparison with that attributed to UA. As shown in Figure 5, except for AA, the potential interferents had little influence on the fluorescence of the AgNPs/N-CQDs, thereby indicating the selectivity of this system for UA detection.

### 2.3. Fluorescence Response of the AgNPs/N-CQDs to Uric Acid

As shown in Figure 6A,B, the fluorescence emission of the AgNPs/N-CQDs responds well to H_2_O_2_, with a detection limit of 0.32 μmol/L being calculated. Subsequently, the suitability of the AgNPs/N-CQDs system to act as a probe for UA was evaluated to determine the response range and detection limit. As shown in Figure 6C,D, the fluorescence intensity of the AgNPs/N-CQDs system was enhanced with an increase in the UA concentration. Over a UA concentration range of 2.0–60 μmol/L, the fluorescence intensity of the AgNPs/N-CQDs correlated well with the UA concentration, giving a linear relationship as follows:*I*/*I*_0_ = 2.53 × 10^−3^*C*_UA_ + 1
where *C_UA_* is the UA concentration, and *I* and *I*_0_ (see Figure 6D) denote the fluorescence intensity of the AgNPs/N-CQDs in the presence and absence of UA, respectively. Thus, a good correlation coefficient (*R*^2^) of 0.9971 was obtained in addition to a UA detection limit of 0.53 μmol/L, and the relative standard deviation (RSD, *n* = 5) was <1.2% at a UA concentration of 10.0 μmol/L, which is comparable with the response ranges and UA limits of detection (LODs) detected reported for previous fluorescence methods [30,31,32,33,34,35,36,37] (Table 1), thereby revealing that our probe based on AgNPs/N-CQDs exhibits an outstanding performance.

### 2.4. Biological Sample Analysis

To further investigate the application and reliability of the developed method in real samples, serum samples from 10 healthy candidates and 10 gout patients were employed for detection (see Table 2). The concentration of the serum samples ranged from 210 to 412 μM, while the concentration of gout serum samples was between 502 and 798 μM. Subsequently, the UA concentrations of the gout serum samples were further confirmed and compared with those of biochemical analyses carried out at the Qianxinan Prefecture People’s Hospital. As shown in Table 2 and Figure 7, an excellent agreement was found between the developed method and the hospital-based biochemical analysis (*p* = 0.79), further demonstrating the reliability and practicality of the AgNPs/N-CQDs probe system. Moreover, a spiked recovery assay was performed on another healthy human serum sample (Table 3), wherein recoveries of 96.3–107% were obtained, with an RSD < 3.7%. The above results clearly demonstrate that the developed probe method is a feasible, highly selective, and highly sensitive system for the detection of UA in complex serum samples, and so could be useful for diagnostic and therapeutic applications.

## 3. Materials and Methods

### 3.1. Chemicals and Reagents

All chemicals and reagents supplied in this work were of analytical grade and were used without further purification. Hydroxyethyl cellulose, L-citrulline, hydrogen peroxide (H_2_O_2_, 30 wt.%), dimethyl sulfoxide, AgNO_3_, NaH_2_PO_4_, Na_2_HPO_4_, NaOH, NaCl, FeCl_3_·6H_2_O, CaCl_2_, MgCl_2_, KCl, glucose, L-tyrosine, L-lysine, L-pll, and ascorbic acid (AA) were purchased from Aladdin Chemistry Co., Ltd. (Shanghai, China). UA and urate oxidase were purchased from Yiji Industries (Shanghai, China). Phosphate buffered saline (PBS) was supplied by Kejin Biological Technology Reagent Co., Ltd. (Wuhan, China).

### 3.2. Measurements

The fluorescence spectra were recorded using an RF-6000 spectrofluorometer (Shimadzu, Japan). TEM (Hitachi-F20 Tokyo, Japan) observations were used to characterise the surface morphology of the prepared AgNPs/N-CQDs composite. XPS was carried out using an ESCALAB 250Xi spectrometer (Thermo Fisher Scientific, Madison, WI, USA). FTIR spectroscopy was performed on a Bruker Nicolet 6700 spectrometer (Karlsruhe, Germany), while the UV−vis absorption spectra were obtained using the Thermo Multiskan Ascent spectrophotometer (Shanghai, China).

### 3.3. Synthesis of the AgNPs/N-CQDs Composite

The AgNPs/N-CQDs composite was synthesised using an aqueous AgNO_3_ solution and employing N-CQDs as both a growing template and a reducing agent. More specifically, hydroxyethyl cellulose (4.00 g) and L-citrulline (1.00 g) were mixed with ultrapure water (60 mL) and subjected to ultrasonication for 15 min. After this time, the above mixture was transferred to a 100 mL Teflon-lined autoclave and heated at 180 °C for 12 h, prior to cooling to 25 ± 15 °C. The resulting solution was then subjected to centrifugation at 12,000 rpm for 10 min, and the majority of stubborn particles were removed. Subsequently, the supernatant was further purified using a 0.22 μm microfilter membrane. Finally, the N-CQDs solution (10 mL, 2000 μg/mL) was adjusted to pH 9.5 using a 0.1 M aqueous NaOH solution, mixed with the AgNO_3_ solution (10 mL, 0.5 mM), stirred for 1 h, and heated at 80 °C for another 30 min to obtain the desired AgNPs/N-CQDs composite.

### 3.4. Detection of Uric Acid

For the purpose of this study, the UA concentration was measured based on the amount of enzymatically generated H_2_O_2_. Different concentrations of H_2_O_2_ (0–90 μmol/L) were added to the AgNPs/N-CQDs and allowed to equilibrate over 45 min, and UA detection was performed in two steps. Initially, urate oxidase (30 μL, 100 μg/mL) was reacted with UA (150 μL, 0–60 μmol/L) in PBS (pH 7.4, 0.1 mol/L) at 37 °C for 45 min in the dark, and then the above reaction solution was directly added to the AgNPs/N-CQDs solution (3.00 mL, 2000 μg/mL) and allowed to equilibrate for 45 min. The fluorescence intensity was detected at an excitation/emission wavelength of 340/408 nm.

To confirm that the AgNPs/N-CQDs can efficiently detect UA in biological samples, serum samples were collected from 10 healthy candidates and 10 gout patients of the Qianxinan Prefecture People’s Hospital. For the UA probe, an aliquot of each serum sample (10 μL) was diluted to 150 μL using a freshly prepared urate oxidase solution (30 µL, 100 μg/mL) and reacted at 37 °C for 45 min in the dark. Subsequently, the reaction solution was directly added to the AgNPs/N-CQDs solution (3.00 mL, 2000 μg/mL) and allowed to equilibrate for 45 min. The fluorescence intensities of the samples were then measured, and the recoveries of UA were calculated using an excitation/emission wavelength of 340/408 nm.

## 4. Conclusions

We herein reported the successful preparation of novel composites of silver nanoparticles (AgNPs) wrapped with nitrogen-doped carbon quantum dots (N-CQDs), which exhibited stable fluorescence properties and an excellent fluorescence performance. The AgNPs/N-CQDs composite was found to be highly selective for the detection of uric acid (UA) based on an enzymatic reaction involving the direct reduction in Ag^+^ by the N-CQDs. Subsequently, a simple, rapid, and effective probe method based on the AgNPs/N-CQDs was developed for the determination of UA analysis. This probe method exhibited a low detection limit of 0.53 μmol/L, and it was successfully applied for the quantitative detection of UA in real serum samples from 10 healthy candidates and 10 gout patients. This method provided several distinct advantages over previous techniques. More specifically, the interference was eliminated in complex matrices due to the highly selective nature of the AgNPs/N-CQDs probe. In addition, this method can be used for the facile and sensitive detection of trace UA present in human blood serum. Moreover, the measurement process is fast, flexible, and simple. These results therefore indicate that bioprobe platforms based on AgNPs/N-CQDs are promising candidates for the efficient determination of trace UA levels, and so could assist disease diagnoses.

## Data Availability

All the data used in this study are available within this article. Further inquiries can be directed to the authors.

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
