# Peer review of "Silver Nanoparticle-Functionalised Nitrogen-Doped Carbon Quantum Dots for the Highly Efficient Determination of Uric Acid"

_molecules, 2022, doi:10.3390/molecules27144586_

Round 1

Reviewer 1 Report

I appreciate that some changes have been made to the manuscript, especially with the characterization of the AgNP/N-CQDs. However, the main problems of the work mentioned in the previous version are still not adequately addressed to consider an improvement in the manuscript. In my opinion, it is necessary to improve the meaning and content of the work to show its originality to be published. I insist that the authors analyze the results of their research and assess whether they are sufficient for this communication.

Author Response

Q1: I appreciate that some changes have been made to the manuscript, especially with the characterization of the AgNP/N-CQDs. However, the main problems of the work mentioned in the previous version are still not adequately addressed to consider an improvement in the manuscript. In my opinion, it is necessary to improve the meaning and content of the work to show its originality to be published. I insist that the authors analyze the results of their research and assess whether they are sufficient for this communication.

Response: Thanks for your constructive suggestion. We revised the last paragraph of the introduction in the manuscript. The content is as follow:

Thus, we herein report the development of a novel AgNPs/N-CQDs nanocomposite as a fluorescent probe for the detection of UA. In particular, the hydroxyethyl cellulose and L-citrulline were firstly used to synthesize the N-CQDs via a hydrothermal method. Because the hydroxyethyl cellulose has abundant -OH functional groups, the as-prepared N-CQDs are facilely coordinating with silver ions. Moreover, the AgNPs present in the AgNPs/N-CQDs will be oxidised to Ag+ by H2O2 to reduce absorption of the light emitted from the N-CQDs and enhance the fluorescence intensity of the nanocomposite. A novel detection method based on the H2O2-mediated fluorescence enhancement of the AgNPs/N-CQDs is then developed to detect UA using the enzymatic catalytic oxidation reaction, and this method is applied to the UA detection of human serum.  

But we do apologize and still do not clear how to improve the content of the research. Could you give a detail explanation about your question? We will very pleasure for your advices, and we would like to revise the manuscript under your suggestion.

Q2: Moderate English changes required

Response: Thank you for the careful reading of our manuscript. We have carefully checked the manuscript for language problems and corrected them.

Reviewer 2 Report

Thank you for your changes.

Author Response

English language and style are fine/minor spell check required

Response: Thank you for the careful reading of our manuscript. We have carefully checked the manuscript for language problems and corrected them.

Reviewer 3 Report

The ms describes the synthesis of Ag Nanoparticle-Functionalised Nitrogen-Doped Carbon Quantum Dots which were utilised for detection of uric acid using fluorescence as analytical methods. Overall, the ms is interesting, however, some revision is needed as follows.

1) there are some typos across the ms that needs to be checked

2) In section 2.2. ascorbic acid should be added as interference species as its concentration is not negligible in real samples

3) in Table 1, the third row "linger" should read instead "linear range"

4) I wonder if the stability of the enzyme is an issue in the fluorescence detection 

Author Response

1) there are some typos across the ms that needs to be checked

Response: Thanks for your comments on our paper. we have carefully checked and corrected them.

2) In section 2.2. ascorbic acid should be added as interference species as its concentration is not negligible in real samples

Response: As shown in Figure 5B of the manuscript. We added the ascorbic acid into the AgNPs/N-CQDs system using the same method. The fluorescence intensity of AgNPs/N-CQDs has slightly changed, the quenching rate of fluorescence is 5.63% ((15070-14221)/15070*100=5.63%) at 5000 μM, meanwhile, the quenching rate of fluorescence is only 0.518% ((15070-14992)/15070*100=0.518%) at 500 μM, the concentration of the ascorbic acid ranged from 16.4 to 64.4 μM in the human serum samples (Bernard Herbeth1,Jean-Claude Guilland, Luc Rochette, e t al. Genetic and environmental contributions to serum ascorbic acid concentrations: the Stanislas Family Study, British Journal of Nutrition (2006), 96, 1013–1020; Leah E Cahill and Ahmed El-Sohemy. Haptoglobin genotype modifies the association between dietary vitamin C and serum ascorbic acid deficiency, American Society for Nutrition, 2010, 92:1494–500), it is very low and the interference of the ascorbic acid is limited, therefore, these results indicate that developed probe method based on AgNPs/N-CQDs are accurate and reliable for the efficient determination of trace UA levels. On the other hand, the UA concentrations of the gout serum samples were further confirmed and compared with those of biochemical analyses carried out at the Qianxinan Prefecture People’s Hospital.

Figure 5. (A) The fluorescence emission spectrum, and (B) the corresponding fluorescence changes of the AgNPs/N-CQDs system upon the addition of 200 μmol/L of UA and 5000 μmol/L of each interfering substance.

3) in Table 1, the third row "linger" should read instead "linear range"

Response: We really appreciate the strict suggestion and the words “Linger” in the manuscript have been revised to “Linger range”.

4) I wonder if the stability of the enzyme is an issue in the fluorescence detection

Response: As shown in the Figure 1 below, the fluorescence intensity of AgNPs/N-CQDs were detected after adding the H2O2, UA and uricase, UA, respectively. There are no obvious fluorescence changes after adding the UA than the black line. This indicate that there is no interaction between AgNPs/N-CQDs and UA. Oppositely, the fluorescence intensity of AgNPs/N-CQDs increased dramatically after adding the UA and uricase, UA, respectively. This shows that the UA was catalyzed by the uricase to generate H2O2 and the H2O2 interacted with the AgNPs/N-CQDs to enhance the fluorescence intensity of AgNPs/N-CQDs. In addition, the previous literature reported the similar analytical approach incubating the uricase with uric acid in phosphate buffer solution at 37 °C for 45 min in the dark to generate H2O2 (Wang X, Zhu G, Liu Z, et al. A novel ratiometric fluorescent probe for the detection of uric acid in human blood based on H2O2-mediated fluorescence quenching of gold/silver nanoclusters[J]. Talanta, 2019, 191: 46-53.). The analytical approach obtained excellent outcomes. In consequence, above results demonstrate that the uricase could work normally under the AgNPs/N-CQDs system.

Figure 1. The fluorescence intensity of AgNPs/N-CQDs solution with 200 μM of H2O2, UA and uricase, UA, respectively. The black line is the blank control.

Round 2

Reviewer 1 Report

Thanks for your updates

This manuscript is a resubmission of an earlier submission. The following is a list of the peer review reports and author responses from that submission.

Round 1

Reviewer 1 Report

Comments and Suggestions for Authors

- Abstract. Could you please highlight the uniqueness of your study here to make the abstract more attractive? Learn more about the results you get and the conclusions you write.

- Introduction that serves here as a theoretical background of the study is well developed but lacks, in my opinion, depth as it is rather descriptive. What could be helpful is adding a table where the recent key results are summarized. Please expand here.

- I think that authors should be more precise about the aim of the study right in the Introduction. There are some places where the aim of the study is indicated, but the readers should be aware of what to expect sooner. Please enlarge the figures and improve resolution to understand better.

-  I´m lacking a discussion section separate from the achieved results. The results should be mirrored here with international literature and the most recent findings.

- The conclusion is descriptive, which is good, but I would expect to read more about consequences, i.e., how the proposed solution might be helpful. In such kind of study, I also think some space should be devoted to factors that affect the results. Could you please expand on these limitations of your research (concerning methodology, settings, inputs, etc.) in the concluding part?

- The list of references should be expanded to cover the recent debate better.

Author Response

Dear editor and reviewers:

Thank you very much for your kind feedback and the reviewers’ valuable suggestions for our manuscript entitled Silver Nanoparticle-Functionalised Nitrogen-Doped Carbon Quantum Dots for the Highly Efficient Determination of Uric Acid (No: 1613915). Those comments are very helpful for revising and improving our paper, as well as the important guiding significance to our researches. Accordingly, we made every attempt to incorporate all the editor and reviewers’ suggestions as thoroughly as possible. And the revised portions have been marked up using the“Track Chang” function in the revised manuscript. We sincerely hope this manuscript will be finally acceptable to be published on Molecules.

Response to Reviewer #1:

  1. - Abstract. Could you please highlight the uniqueness of your study here to make the abstract more attractive? Learn more about the results you get and the conclusions you write.

Response: We really appreciate the reviewer’s constructive suggestion and we do agree with the reviewer that there should be some narrative to make the abstract more attractive. Thus, some words have been added in the “Abstract” section of the revised manuscript to summarize the consequences in the abstract as follows: The results are consistent with those of the hospital. The AgNP/N-CQD system can provide a universal detection platform for multispecies ratio fluorescence of H2O2 generation in other biological systems.

  1. - Introduction that serves here as a theoretical background of the study is well developed but lacks, in my opinion, depth as it is rather descriptive. What could be helpful is adding a table where the recent key results are summarized. Please expand here.

Response: Thanks for your comments on our paper. We try our best to revise our paper according to your suggestion. We do apologize it is difficulty to our further perfect in research paper with table form.

  1. I think that authors should be more precise about the aim of the study right in the Introduction. There are some places where the aim of the study is indicated, but the readers should be aware of what to expect sooner. Please enlarge the figures and improve resolution to understand better.

Response: Thanks to the valuable suggestion. We have revised our paper according to your suggestion. All of the images had been improved according the requirement.

  1. - I´m lacking a discussion section separate from the achieved results. The results should be mirrored here with international literature and the most recent findings.

Response: We appreciate the professional suggestion, the comparing the results of the fluorescence method with the other previously reported methods are presented in Table 1.

Table 1. Comparison of developed method with others in terms of response range and LOD

Method

Probe

Linger range (μM)

LOD

(μM)

Ref.

Fluorescence

Ag-CQDs

0.005-100

3.5×10-4

[29]

Fluorescence

Eu-BDC@FM

0-200

0.6

[30]

Fluorescence

N-CDs

0.5-150

0.06

[31]

Fluorescence

TPE@SNW-1

10.0-150

4.94

[32]

Fluorescence

Au/Ag NCs

5.0-50

5.1

[33]

Fluorescence

N,Co-CDs

0.01-100

3.4×10-3

[34]

Fluorescence

SiNPs

10.0-800

0.75

[35]

Fluorescence

luminol-Tb NPs

0.1-50

2.8×10-2

[36]

Fluorescence

AgNPs/N-CQDs

2.0-60

0.53

This work

  1. - The conclusion is descriptive, which is good, but I would expect to read more about consequences, i.e., how the proposed solution might be helpful. In such kind of study, I also think some space should be devoted to factors that affect the results. Could you please expand on these limitations of your research (concerning methodology, settings, inputs, etc.) in the concluding part?

Response: Thanks for your comments on our paper. We did the best revised our paper according to your suggestion.

Reviewer 2 Report

Nice work. The manuscript with title "Silver Nanoparticle-Functionalised Nitrogen-Doped Carbon Quantum Dots for the Highly Efficient Determination of Uric Acid" deals with an interesting issue, that of the detection of uric acid (UA) using  silver nanoparticles (AgNPs) wrapped with nitrogen-doped carbon quantum dots (N-CQDs).

The authors synthesize and characterize the silver nanoparticles wrapped with carbon quantum dots. Then these composites were used  for 
the detection of uric acid using the enzymatic reaction of uric acid oxidase. 

I would like to draw the attention to a few points.

Regarding the XRD results please provide the JCPDS card.

Regarding the N-CQDs please provide  TEM images which support the manuscript. 

The image you have shows two Ag particles.  The one of them is over 30nm but you write that they have size from 10 to 14. Please provide images which shows many Ag particles. 

Author Response

Dear editor and reviewers:

Thank you very much for your kind feedback and the reviewers’ valuable suggestions for our manuscript entitled Silver Nanoparticle-Functionalised Nitrogen-Doped Carbon Quantum Dots for the Highly Efficient Determination of Uric Acid (No: 1613915). Those comments are very helpful for revising and improving our paper, as well as the important guiding significance to our researches. Accordingly, we made every attempt to incorporate all the editor and reviewers’ suggestions as thoroughly as possible. And the revised portions have been marked up using the“Track Chang” function in the revised manuscript. We sincerely hope this manuscript will be finally acceptable to be published on Molecules.

Response to Reviewer #2:

Comments and Suggestions for Authors

Nice work. The manuscript with title "Silver Nanoparticle-Functionalised Nitrogen-Doped Carbon Quantum Dots for the Highly Efficient Determination of Uric Acid" deals with an interesting issue, that of the detection of uric acid (UA) using silver nanoparticles (AgNPs) wrapped with nitrogen-doped carbon quantum dots (N-CQDs). The authors synthesize and characterize the silver nanoparticles wrapped with carbon quantum dots. Then these composites were used for the detection of uric acid using the enzymatic reaction of uric acid oxidase. I would like to draw the attention to a few points.

  1. Regarding the XRD results please provide the JCPDS card.

Response: We very much appreciate for the careful reading of our manuscript. This is very important for improving the quality of our manuscript, and the corresponding PDF cards have been provided and revised in the manuscript (Figure 1B).

Figure 1. (B) XRD spectrum of the AgNP/N-CQD composite.

  1. Regarding the N-CQDs please provide TEM images which support the manuscript.

Response: Thanks for the helpful suggestion. We added the TEM images of the N-CQD in the revised manuscript. The TEM image (inset in Figure 1A) clearly showed N-CQD with spherical morphology, diameter of 3.6-5.2 nm, and the particles are amorphous carbon without obvious lattice fringes.

Figure 1. (A) TEM image of the AgNP/N-CQD composite. Inset shows the TEM image of the N-CQD.

  1. The image you have shows two Ag particles. The one of them is over 30nm but you write that they have size from 10 to 14. Please provide images which shows many Ag particles.

Response: We very much appreciate for the careful reading of our manuscript. As shown in Figure R1 below, the size of the most Ag particles is about 10-14 nm. But these Ag nanoparticles tend to agglomerate causing the large size than the normal Ag nanoparticles. Therefore, we only selected a part of the picture to put in the manuscript.

Figure R1. The TEM image of the Ag particles.

Round 2

Reviewer 1 Report

I appreciate that some changes were made to the manuscript. However, some issues were justified extensively in the reviewers' responses but not included as improvements in the manuscript. Most of the previous suggestions were partially heeded, so I cannot issue a favorable recommendation. I consider it prudent for the authors to analyze whether their research results are sufficient for this communication. In my opinion, it is necessary to improve the meaning and content of the work to show its originality to be published.

Reviewer 2 Report

Comments:

"Response: Thanks for the helpful suggestion. We added the TEM images of the N-CQD in the revised manuscript. The TEM image (inset in Figure 1A) clearly showed N-CQD with spherical morphology, diameter of 3.6-5.2 nm, and the particles are amorphous carbon without obvious lattice fringes".

Two of three particles shown (inset in Figure 1A) have fringes and judging from Figure R1. (Response to reviewer’s comments)  and their contrast these particles are possibly Ag particles. There isn't any evidence of carbon Quantum dots.

"Response: We very much appreciate for the careful reading of our manuscript. As shown in Figure R1 below, the size of the most Ag particles is about 10-14 nm. But these Ag nanoparticles tend to agglomerate causing the large size than the normal Ag nanoparticles. Therefore, we only selected a part of the picture to put in the manuscript."

 Agglomeration (obvious in the image) does not result in larger sizes of these Ag particles, particles are "spherical" and distinct even when they are touching or partly overlaping with eachother. Their size is by far larger than 14 nm.